# DISCRETE PREDICTOR-CORRECTOR DIFFUSION MODELS FOR IMAGE SYNTHESIS

**José Lezama**[1]*, **Tim Salimans**[1], **Lu Jiang**[1], **Huiwen Chang**[1], **Jonathan Ho**[1], **Irfan Essa**[1,2]

[1] Google Research  [2] Georgia Institute of Technology

## ABSTRACT

We introduce Discrete Predictor-Corrector diffusion models (DPC), extending predictor-corrector samplers in Gaussian diffusion models to the discrete case. Predictor-corrector samplers are a class of samplers for diffusion models, which improve on ancestral samplers by correcting the sampling distribution of intermediate diffusion states using MCMC methods. In DPC, the Langevin corrector, which does not have a direct counterpart in discrete space, is replaced with a discrete MCMC transition defined by a learned corrector kernel. The corrector kernel is trained to make the correction steps achieve asymptotic convergence, in distribution, to the correct marginal of the intermediate diffusion states. Equipped with DPC, we revisit recent transformer-based non-autoregressive generative models through the lens of discrete diffusion, and find that DPC can alleviate the compounding decoding error due to the parallel sampling of visual tokens. Our experiments show that DPC improves upon existing discrete latent space models for class-conditional image generation on ImageNet, and outperforms continuous diffusion models and GANs, according to standard metrics and user preference studies.

## 1 INTRODUCTION

Generative Adversarial Networks (GANs) are the leading model class for a wide variety of content creation tasks (Goodfellow et al., 2014; Brock et al., 2018; Karras et al., 2020). Recently, however, likelihood-based models, such as diffusion models (Dhariwal & Nichol, 2021; Ho et al., 2020; 2022) and generative transformers (Ramesh et al., 2021; Esser et al., 2021b; Chang et al., 2022), have started rivaling GANs in offering an alternative training paradigm with superior training stability and improved generation diversity. In particular, ADM (Dhariwal & Nichol, 2021) and CDM (Ho et al., 2022) presented diffusion models attaining better perceptual quality on the class-conditional ImageNet benchmark compared to BigGAN (Brock et al., 2018).

Sampling speed, however, is still a bottleneck hindering the practical application of diffusion models. These models can be orders of magnitude slower than GANs, due to the need to take up to hundreds of steps to synthesize a single image during inference. Recently, *discrete diffusion* has been receiving attention as a promising direction for achieving an improved trade-off between generation quality and efficiency. Like the continuous (Gaussian) diffusion process (Sohl-Dickstein et al., 2015; Song & Ermon, 2019; Ho et al., 2020; Gu et al., 2022), these models incrementally corrupt training data until a known base distribution is reached, and this corruption process is reversed when sampling from the learned model. Unlike continuous diffusion models, the corruption is applied in a latent, possibly low-dimensional, discrete space. The image generation quality of discrete diffusion models is still inferior to that of continous diffusion models. For example, the state-of-the-art discrete diffusion model (*i.e.*, VQ-Diffusion (Gu et al., 2022)) still notably underperforms CDM (Ho et al., 2022) and BigGAN (Brock et al., 2018) on ImageNet without the guidance from external classifiers.

Contemporarily, *non-autoregressive transformers* (Chang et al., 2022; Gu et al., 2022; Zhang et al., 2021; Lezama et al., 2022) have demonstrated promising performances in both perceptual image quality and efficiency on the ImageNet benchmark. In particular, a non-autoregressive transformer model named MaskGIT (Chang et al., 2022) achieves comparable generation quality to the leading diffusion model ADM on ImageNet, while enjoying two orders-of-magnitude faster inference speed.

---

*Corresponding author `joselezama@google.com`

It brings down the generation time of ADM from $\sim$500 steps to only 18 steps, making it possible to generate an image within 0.1 second on a TPU device. Non-autoregressive transformers are trained to predict masked visual tokens, inspired by masked language models such as BERT (Devlin et al., 2019). The decoding process, similar to non-autoregressive sampling techniques from machine translation (Ghazvininejad et al., 2019; Kong et al., 2020), predicts all missing tokens in parallel, starting from a fully masked sequence, and subsequently following an iterative refinement schedule. By viewing non-autoregressive transformers through the lens of discrete diffusion, our analysis yields new insights into a critical issue with these models, namely the *compounding decoding error*, which causes a mismatch between the inference and training distributions of the intermediate latents produced during the parallel sampling process.

To tackle the *compounding decoding error*, we propose Discrete Predictor-Corrector (DPC) diffusion models that introduce iterative refinement of the intermediate diffusion states. DPC learns a corrector kernel that, coupled with the reverse diffusion predictor, forms a discrete Markov Chain Monte Carlo (MCMC) predictor-corrector algorithm that has the correct marginal distribution of the intermediate latents as its limiting distribution. The proposed DPC model is a new type of discrete diffusion model that is distinct from both non-autoregressive transformers and conventional discrete diffusion models. Compared to non-autoregressive transformers (Chang et al., 2022; Lezama et al., 2022; Gu et al., 2022), DPC introduces new techniques to perform multi-step correction of intermediate states in the sampling process. Furthermore, it provides a theoretical underpinning for this model class, and shows that, in the ideal case, it can completely correct the train/test mismatch between the distributions of intermediate states. DPC advances conventional discrete diffusion by introducing a new discrete MCMC correction kernel, which can be considered as a discrete analogue to the Langevin corrector that was of (Song et al., 2021) for the continuous case.

We empirically validate that DPC is able to achieve a good quality-vs-efficiency trade-off on two tasks: class-conditional image generation on the ImageNet dataset and unconditional generation on the Places2 (Zhou et al., 2017) dataset. The results show that DPC performs favorably against both non-autoregressive transformers (Chang et al., 2022; Lezama et al., 2022; Esser et al., 2021b) and discrete diffusion baselines (Gu et al., 2022) while maintaining a fast inference speed. In particular, without the help of external classifiers, DPC outperforms the state-of-the-art continuous diffusion models (*i.e.*, ADM (Dhariwal & Nichol, 2021) and CDM (Ho et al., 2022)) in FID (Heusel et al., 2017), thereby establishing a new state-of-the-art on the high-resolution (512×512) image synthesis task on ImageNet. When leveraging an external pre-trained classifier and upsampling, DPC produces state-of-the-art class-conditional generation, yielding better Inception Score (IS) and FID compared to state-of-the-art GANs, (e.g. StyleGAN-XL (Sauer et al., 2022)) and continuous diffusion models (e.g. ADM (Dhariwal & Nichol, 2021)). Furthermore, we present user preference studies that confirm the perceptual quality provided by DPC.

## 2    BACKGROUND AND PROBLEM STATEMENT

### 2.1    DISCRETE DIFFUSION MODELS

Let $\boldsymbol{x}_0 \in \{1, \ldots, K\}^N$ be a vector of discrete data, such as an image or a set of image tokens obtained by Vector-Quantized (VQ) encoding with a dictionary of $K$ elements. Using a discrete diffusion process $q(\boldsymbol{x}_{t+1}|\boldsymbol{x}_t)$, we can sample a sequence of latent variables $\boldsymbol{x}_1, \boldsymbol{x}_2, \ldots, \boldsymbol{x}_T$ such that the final latent $\boldsymbol{x}_T$ has a simple known and fixed distribution $p(\boldsymbol{x}_T)$. Starting from a sample from $p(\boldsymbol{x}_T)$, we can then reverse this process using a learned reverse diffusion model $p_\theta(\boldsymbol{x}_{t-1}|\boldsymbol{x}_t)$, eventually producing a sample $\boldsymbol{x}_0$.

One instance of such a discrete diffusion process $q$, is the *absorbing state diffusion process* from (Austin et al., 2021), in which we set $\boldsymbol{x}_t = \boldsymbol{x}_0 \odot \boldsymbol{m}_t$, where $\boldsymbol{m}_t$ is a vector of binary masks that starts out as all ones, $\boldsymbol{m}_0 = \boldsymbol{1}$, and ends up as all zeros, $\boldsymbol{m}_T = \boldsymbol{0}$. In between we gradually evolve $\boldsymbol{m}_t$ by randomly setting more and more of its elements to zero according to $\boldsymbol{m}_t \sim q(\boldsymbol{m}_t|\boldsymbol{m}_{t-1})$.

The reverse diffusion model $p_\theta(\boldsymbol{x}_{t-1}|\boldsymbol{x}_t, \boldsymbol{m}_t)$ can be constructed by first sampling a new mask $q(\boldsymbol{m}_{t-1}|\boldsymbol{m}_t)$, and then sampling a new value for $\boldsymbol{x}_{t-1}$ corresponding to those values of $\boldsymbol{m}_{t-1}$ that are newly unmasked, *i.e.* sampling $x_{t-1}^i \sim p_\theta(x_{t-1}^i|\boldsymbol{x}_t)$ for those $i$ for which $m_{t-1}^i = 1$, and $m_t^i = 0$.

## 2.2 ORDER-AGNOSTIC AUTOREGRESSIVE MODELS

When sampling from a discrete diffusion model with a sufficiently large number of steps $t$, only a single element of $\boldsymbol{x}$ is masked/unmasked in each transition, so we only need to sample a single $x_{t-1}^i$ at each step of the reverse diffusion process. In this case, $p_\theta$ is an *order-agnostic autoregressive model*, as introduced by Uria et al. (2014) and discussed in more detail by Hoogeboom et al. (2022). As shown in Uria et al. (2014), these models can be trained efficiently by maximizing the likelihood

$$\sum_{i \text{ s.t. } m_t^i = 0} \log p_\theta(x_{t-1}^i = x_0^i | \boldsymbol{x}_t, \boldsymbol{m}_t), \tag{1}$$

where the sum is over all possible orderings of the data that are consistent with $\boldsymbol{m}_t$: this has lower variance than just maximizing the likelihood for the single ordering that was actually sampled by $q(\cdot)$, but is otherwise equivalent. Alternatively, it can be interpreted as denoising the corrupted data $\boldsymbol{x}_t$ towards the original clean data $\boldsymbol{x}_0$, trained by maximizing the log-likelihood $\log \tilde{p}_\theta(\boldsymbol{x}_0 | \boldsymbol{x}_t, \boldsymbol{m}_t)$ for the whole vector $\boldsymbol{x}_0$ at once, under a one-step model $\tilde{p}_\theta$ defined by

$$\tilde{p}_\theta(\boldsymbol{x}_0 | \boldsymbol{x}_t, \boldsymbol{m}_t) = \prod_i p_\theta(x_0^i | \boldsymbol{x}_t, \boldsymbol{m}_t), \tag{2}$$

where we use the $\tilde{p}_\theta$ notation to distinguish this factorized one-step denoising model from the full multi-step generative model $p_\theta(\boldsymbol{x}_0) = \sum_{\boldsymbol{x}_{t>0}} \prod_t p_\theta(\boldsymbol{x}_{t-1} | \boldsymbol{x}_t)$. In the following we use $\tilde{p}_\theta$ for all distributions affected by using the one-step model in the generative process.

## 2.3 NON-AUTOREGRESSIVE TRANSFORMERS FOR IMAGE SYNTHESIS

Transformer-based models generate images in two stages (Esser et al., 2021b; Ramesh et al., 2021). First, the image is quantized into a grid of discrete tokens by a VQ-autoencoder (van den Oord et al., 2017). In the second stage, an autoregressive transformer decoder (Vaswani et al., 2017; Chen et al., 2020a) is learned on the flattened token sequence to generate image tokens sequentially using autoregressive decoding. In the end, the generated codes are mapped to pixel space using the VQ-decoder learned in the first stage (Esser et al., 2021b).

Recently, non-autoregressive transformers (Chang et al., 2022; Gu et al., 2022; Zhang et al., 2021) are proposed to improve the second stage, adapted from non-autoregressive machine translation (Ghazvininejad et al., 2019; Kong et al., 2020). An example of such model is the Masked Generative Image Transformer (MaskGIT) (Chang et al., 2022), which follows the masked modeling in BERT (Devlin et al., 2019) (equation 2). During decoding, MaskGIT starts with all the image tokens masked out. In each inference step, it uses parallel decoding, *i.e.* predicting all tokens simultaneously while only keeping the ones with the highest prediction scores. The remaining tokens are masked out and will be re-predicted in the next iteration. The mask ratio is made decreasing, according to a cosine function, until all tokens are generated within a few iterations of refinement.

## 2.4 COMPOUNDING DECODING ERROR IN NON-AUTOREGRESSIVE TRANSFORMERS

To sample from an autoregressive model, one would typically sample one element $i$ at a time, which is costly when the total number of elements $N$ in $\boldsymbol{x}$ is large. By ignoring the dependencies between the $x_{t-1}^i$, one can accelerate this process by sampling several $x_{t-1}^i$ in parallel at each transition. This idea has been explored in discrete diffusion models by (Gu et al., 2022; Hoogeboom et al., 2022), and is exploited by non-autoregressive generative transformers (Chang et al., 2022). As suggested in prior work (Austin et al., 2021; Gu et al., 2022; Savinov et al., 2021; Lezama et al., 2022), there exists a close relation between non-autoregressive transformers and discrete diffusion models. For example, the masked-modeling training of state-of-the-art MaskGIT (Chang et al., 2022), can be modeled by the discrete diffusion process with the absorbing state ([MASK]) (Austin et al., 2021). MaskGIT's parallel sampling may be understood as a reverse diffusion process using the one-step model in (2).

However, parallel sampling introduces errors as the factorized $\tilde{p}_\theta$ does not exactly match the original generative model $p_\theta$ for the joint distribution of the elements of $\boldsymbol{x}$. As a result, the marginal inference distribution $\tilde{p}_\theta(\boldsymbol{x}_t)$ will deviate from the training distribution $q(\boldsymbol{x}_t)$ as sampling progresses. We refer to this as a *compounding decoding error*, as small differences between $q(\boldsymbol{x}_{t-1} | \boldsymbol{x}_t)$ and $\tilde{p}_\theta(\boldsymbol{x}_{t-1} | \boldsymbol{x}_t)$ can accumulate into large differences between $q(\boldsymbol{x}_t)$ and $\tilde{p}_\theta(\boldsymbol{x}_t)$ after many sampling steps (*cf.* teacher forcing error).

## 2.5 PREDICTOR-CORRECTOR SAMPLERS

Predictor-Corrector samplers were proposed by Song et al. (2021) for use in Gaussian diffusion models: before applying each time transition in the generative model $p_\theta(\boldsymbol{x}_{t-1}|\boldsymbol{x}_t)$, these samplers correct the distribution of $\boldsymbol{x}_t$ by applying one or more steps of Langevin MCMC at constant timestep $t$, thereby bringing the distribution of $\boldsymbol{x}_t$ closer to $q(\boldsymbol{x}_t)$. The corrector has the form

$$\boldsymbol{x}_t \leftarrow \boldsymbol{x}_t + \epsilon_t s_\theta(\boldsymbol{x}_t, t) + \sqrt{2\epsilon_t}\boldsymbol{z}, \ \boldsymbol{z} \sim \mathcal{N}(0, I),$$

where $s_\theta(\boldsymbol{x}_t, t) \approx \nabla_{\boldsymbol{x}} \log q_t(\boldsymbol{x}_t)$ is the learned score function and $\epsilon_t$ is a step size.

## 3 DISCRETE PREDICTOR-CORRECTOR SAMPLERS

In this section, we address the critical issue of compounding decoding error due to parallel sampling in non-autoregressive transformers. We propose to mitigate the gap between $q(\boldsymbol{x}_t)$ and $\tilde{p}_\theta(\boldsymbol{x}_t)$ by applying a learned discrete corrector MCMC step to iteratively improve intermediate diffusion states. This allows our models to exploit the computational advantages of parallel sampling in non-autoregressive transformers, while maintaining high synthesis quality.

To reduce the compounding decoding error in $\tilde{p}_\theta(\boldsymbol{x}_t)$, we adjust the sampled $\boldsymbol{x}_t$ to more closely resemble samples from $q(\boldsymbol{x}_t)$, before applying the time-transition from $\boldsymbol{x}_t$ to $\boldsymbol{x}_{t-1}$. In the discrete diffusion case, however, there is no direct counterpart to the score function that gives the rate of change required to improve the likelihood of a sample (Campbell et al., 2022). Instead, we propose to utilize an MCMC corrector step of the following form:

$$\hat{\boldsymbol{x}}_0 \sim \tilde{p}_\theta(\hat{\boldsymbol{x}}_0|\boldsymbol{x}_t, \boldsymbol{m}_t) \tag{3}$$

$$\boldsymbol{x}'_t \sim p_\phi(\boldsymbol{x}_t|\hat{\boldsymbol{x}}_0, t), \tag{4}$$

where $p_\phi$ is a learned corrector distribution[1], trained so that when applying this procedure multiple times, the distribution of $\boldsymbol{x}'_t$ converges to $q(\boldsymbol{x}_t)$, making this a valid corrector kernel (Section 3.1).

Since $\boldsymbol{x}_t = \hat{\boldsymbol{x}}_0 \odot \boldsymbol{m}_t$, the corrector distribution $p_\phi$ can equivalently be stated in terms of the mask $\boldsymbol{m}_t$, which is resampled at every corrector step, $\boldsymbol{m}_t \sim p_\phi(\boldsymbol{m}_t|\hat{\boldsymbol{x}}_0, t)$. The mask is a collection of binary variables, of which a known number $k$ are equal to one at a given time step $t$, and the others are zero. Thus, the corrector distribution $p_\phi(\boldsymbol{m}_t|\hat{\boldsymbol{x}}_0, t)$ is a categorical distribution over all possible binary masks with $k$ non-zero elements. We model this distribution using the *Plackett-Luce model* (Plackett, 1975), which gives

$$p_\phi(\boldsymbol{m}_t|\hat{\boldsymbol{x}}_0, t) = \sum_{\boldsymbol{c}_k(\boldsymbol{m}_t)} \prod_{i=1}^{k} \frac{\exp[l_\phi^{c_k^i}(\hat{\boldsymbol{x}}_0, t)]}{\sum_{j=i}^{k} \exp[l_\phi^{c_k^j}(\hat{\boldsymbol{x}}_0, t)] + \sum_{l:\boldsymbol{m}_t^l=0} \exp[l_\phi^l(\hat{\boldsymbol{x}}_0, t)]}, \tag{5}$$

where $\mathbf{l}_\phi(\hat{\boldsymbol{x}}_0, t)$ is the $N$-dimensional output of a neural network corrector model, with elements $l_\phi^i(\hat{\boldsymbol{x}}_0, t)$ representing the relative log-odds (logits) of $\boldsymbol{m}_t^i$ being positive for the given $\hat{\boldsymbol{x}}_0$, and where we sum over all possible permutations $\boldsymbol{c}_k$ of the indices of the $k$ non-zero elements of the mask $\boldsymbol{m}_t$. In numpy notation: $\boldsymbol{c}_k = $ `np.permute(np.nonzero(`$\boldsymbol{m}_t$`))`. We provide more background on this model in Appendix A.

In practice, we can sample from the mask model by forming perturbed logits by adding i.i.d. Gumbel noise, $\tilde{l}^i = l_\phi^i(\hat{\boldsymbol{x}}_0, t) + G^i$, where $G^i \sim \text{Gumbel}(0, 1)$, and setting the mask to 1 for the top-$k$ elements of $\tilde{l}$, with $k$ determined by $t$. This is known as the Gumbel-top-$k$ trick (Kool et al., 2019).

## 3.1 DISCRETE PREDICTOR-CORRECTOR TRAINING

The learned corrector function $p_\phi$ is trained to achieve *detailed balance* between the MCMC corrector steps (3) and (4), and the training marginal $q(\boldsymbol{x}_t)$. Let $J(\boldsymbol{x}'_t|\boldsymbol{x}_t)$ be the transition kernel of the Markov chain given by (3) and (4). Then, this Markov chain has $q(\boldsymbol{x}_t)$ as stationary distribution if

$$p_\phi(\boldsymbol{x}_t|\hat{\boldsymbol{x}}_0) = \frac{\tilde{p}_\theta(\hat{\boldsymbol{x}}_0|\boldsymbol{x}_t)q(\boldsymbol{x}_t)}{Z(\hat{\boldsymbol{x}}_0)}, \tag{6}$$

---

[1]We use the subscript $\theta$ for models of the reverse process and $\phi$ for the corrector model, which goes in the forward direction. The random variables described by the models are determined by what is inside the brackets.

with $Z(\hat{x}_0) = \sum_{x_t} \tilde{p}_\theta(\hat{x}_0|x_t)q(x_t)$. Assuming (6), the detailed balance condition of $J(x'_t|x_t)$ with respect to $q(x_t)$ follows from:

$$q(x_t)J(x'_t|x_t) = q(x_t) \sum_{\hat{x}_0} p_\phi(x'_t|\hat{x}_0)\tilde{p}_\theta(\hat{x}_0|x_t) \tag{7}$$

$$= q(x_t) \sum_{\hat{x}_0} \left[ \frac{\tilde{p}_\theta(\hat{x}_0|x'_t)q(x'_t)}{Z(\hat{x}_0)} \right] \tilde{p}_\theta(\hat{x}_0|x_t) \tag{8}$$

$$= q(x_t)q(x'_t) \sum_{\hat{x}_0} \frac{\tilde{p}_\theta(\hat{x}_0|x'_t)\tilde{p}_\theta(\hat{x}_0|x_t)}{Z(\hat{x}_0)} = q(x'_t)J(x_t|x'_t), \tag{9}$$

where the last step is derived from applying the same operations in the other direction.

The goal of training the corrector kernel is then to obtain $p_\phi(x_t|\hat{x}_0) \propto \tilde{p}_\theta(\hat{x}_0|x_t)q(x_t)$ (equation 6), which can be done by minimizing the KL divergence

$$KL\big[\tilde{p}_\theta(\hat{x}_0|x_t)q(x_t)\|p_\phi(x_t|\hat{x}_0)Z(\hat{x}_0)\big] = -\mathbb{E}_{(\hat{x}_0,x_t)\sim\tilde{p}_\theta(\hat{x}_0|x_t)q(x_t)} \log p_\phi(x_t|\hat{x}_0) + C, \tag{10}$$

where $C$ are terms that do not include $\phi$. Importantly, the denoiser $\tilde{p}_\theta$ is pre-trained and its parameters are frozen during the training of the corrector $p_\phi$. Thus, the corrector distribution can be trained by first obtaining a masked sample from the training distribution $q(x_t)$, and then using the denoising model $\tilde{p}_\theta(\hat{x}_0|x_t)$ (with frozen weights $\theta$) to obtain a corresponding $\hat{x}_0$, after which $p_\phi(x_t|\hat{x}_0)$ can be trained by maximum likelihood. Since $x_t$ is completely determined by $\hat{x}_0$ and the mask $m_t$, this is equivalent to training $p_\phi$ to maximize the likelihood of the sampled mask $m_t$.

Under our chosen model in (5), evaluating the likelihood for all mask elements jointly would require summing over all possible masks, which is computationally expensive. Another simple training objective is binary cross entropy applied independently to each mask element using $p_\phi(m_t^i = 1|\hat{x}_0, t) \approx \sigma(l_\phi^i(\hat{x}_0, t))$, where $\sigma$ is the sigmoid function. In our experiments we find this simplified loss to be very effective. We discuss this choice in greater depth in Appendix A.

Following MaskGIT, we use the cosine schedule for determining the masking ratio during both training and inference. Let $T$ denote the time horizon. For $S$ sampling iterations, we set the masking ratio to $\gamma(t) = 1 - \cos(t/S \cdot \pi/2)$, for $t \in \{1, \ldots, T/S\}$. During training of both the generator $p_\theta$ and corrector $p_\phi$, we first sample a time $t$, i.e. $t \sim \mathcal{U}_{[0,1]}$, before inputting it to compute $\gamma(t)$. Likewise, during inference, we compute $\gamma(t)$ using the current decoding step. In practice, this scheduling gives more weight to the transitions with lower masking rates, in a similar fashion to the cosine weighing scheme of other diffusion models (e.g., (Hoogeboom et al., 2021)).

## 3.2 Efficient Implementation using Shortcut Time Transitions

Our predictor-corrector sampler consists of taking one predictor step:

$$x_{t-1} \sim p_\theta(x_{t-1}|x_t) \text{ (step 1)},$$

followed by one or multiple corrector steps:

$$\hat{x}_0 \sim \tilde{p}_\theta(\hat{x}_0|x_{t-1}) \text{ (step 2) followed by } x_{t-1} \sim p_\phi(x_{t-1}|\hat{x}_0) \text{ (step 3)}.$$

Under the exact diffusion distribution $q$, we have that

$$\sum_{x_{t-1}} q(x_0|x_{t-1})q(x_{t-1}|x_t) = q(x_0|x_t).$$

Since our model $\tilde{p}_\theta(x_0|x_t)$ is trained to approximate $q(x_0|x_t)$ as well as possible, we then also have

$$\sum_{x_{t-1}} \tilde{p}_\theta(\hat{x}_0|x_{t-1})p_\theta(x_{t-1}|x_t) \approx \tilde{p}_\theta(\hat{x}_0|x_t),$$

to a close approximation. Hence, step 1 and step 2 of our predictor-corrector sampler can be combined into a single step $\hat{x}_0 \sim \tilde{p}_\theta(\hat{x}_0|x_t)$. Using this result, the *shortcut time transition* becomes

$$\hat{x}_0 \sim \tilde{p}_\theta(\hat{x}_0|x_t) \text{ followed by } x_{t-1} \sim p_\phi(x_{t-1}|\hat{x}_0).$$

This avoids one sampling step, and offers a more efficient way of implementing DPC. In the special case of a single corrector step, this implementation of DPC also closely resembles the non-autoregressive transformer Token-Critic (Lezama et al., 2022).

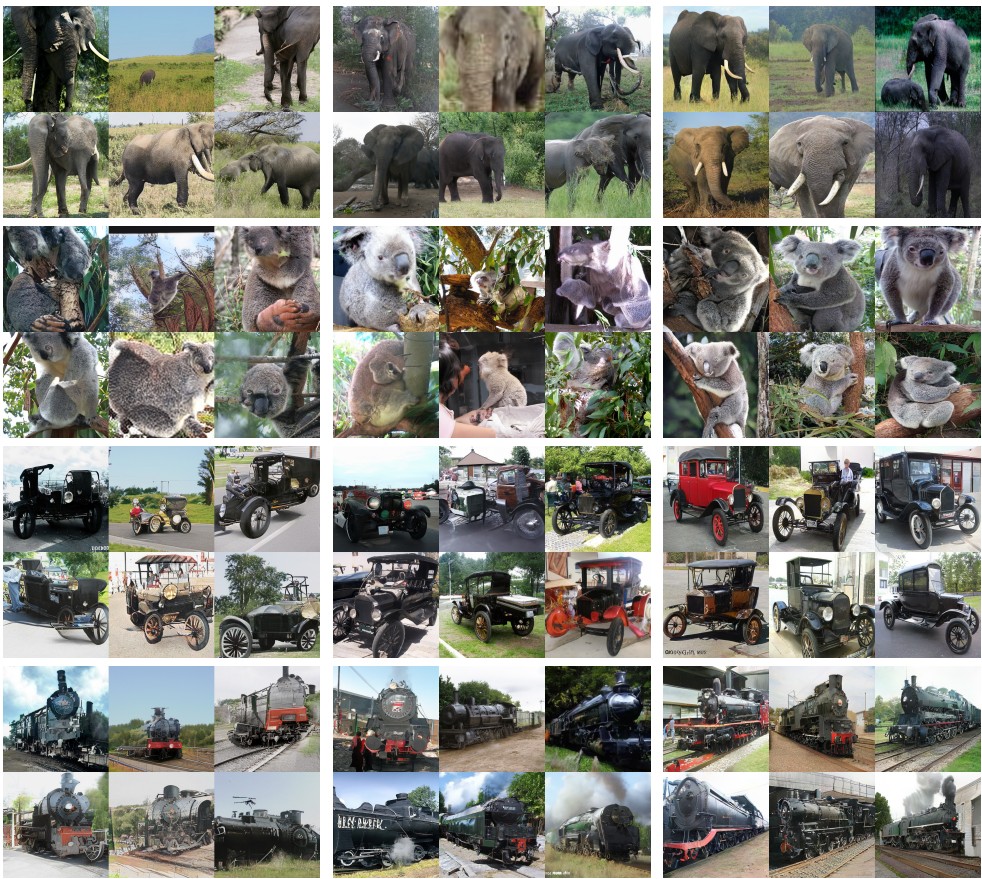

Figure 1: Random samples from ImageNet 512×512 class-conditional generation for selected classes: 'tusker', 'koala', 'model T' and 'steam locomotive'. **Left:** StyleGAN-XL (NFE = 1). **Center:** ADM + Classifier Guidance + Upsampling (NFE = 250 × 3). **Right:** DPC-light(5) (NFE = 66).

## 4 EXPERIMENTS

### 4.1 EXPERIMENTAL SETUPS

**Datasets, Baselines, and Metrics** We evaluate the proposed DPC model for the tasks of class-conditional and unconditional image generation. For class-conditional image generation, we use the standard ImageNet dataset at resolutions 256×256 and 512×512. For unconditional generation, we use the Places2 dataset (Zhou et al., 2017) of 1.8M images at resolution 512×512.

We compare to state-of-the-art models on class-conditional image generation, which include (1) *non-autoregressive generative transformers*: MaskGIT (Chang et al., 2022) and Token-Critic (Lezama et al., 2022); (2) *continuous diffusion models*: ADM (Dhariwal & Nichol, 2021) and CDM (Ho et al., 2022); (3) discrete diffusion model: VQ-Diffusion (Gu et al., 2022); (4) *GAN models*: BigGAN (Brock et al., 2018) and StyleGAN-XL (Sauer et al., 2022). We use FID (Heusel et al., 2017), Inception Score (IS) (Salimans et al., 2016), and Precision vs. Recall (Kynkäänniemi et al., 2019) to evaluate perceptual quality. We also report the number of neural function evaluations (NFE) required by each method. ImageNet generation is evaluated against the training set, and Places2 against the validation set, following Chang et al. (2022). Numbers for the baseline models are quoted from their respective papers except for MaskGIT for which we used a provided model. In addition, we conduct user preference studies to further evaluate the perceptual quality and diversity of the models' samples.

**Refinement Schedule** We evaluate two versions of DPC, varying the number of correction steps $c(t)$ in each intermediate state $t$, where we take $c(t)$ to include the corrector step in the shortcut time transition (section 3.2):

- DPC-full($C$): $c(t) = C$
- DPC-light($C$): $c(t) = \min(s_C(t, \tau_1), s_C(T - t, T - \tau_2))$,

with $s_C(t) = \max(1, C + \min(0, t - \tau))$ a slanted step function, making $c(t)$ a trapezoid with the top vertices at $t = \tau_1$ and $t = \tau_2$. In DPC-light, $c(t)$ concentrates more correction steps around $t = \tau_1$ to $t = \tau_2$. To further reduce the cost, we stop the reverse process at $t = T - 5$. Table 5 presents results varying $C$, and shows that more DPC correction steps effectively lead to improved sampling performance. Ablation studies for $\tau_1$, $\tau_2$ and $\beta$ are included in Appendix B.

**Upsampling in Discrete Latent Space**  Following the success of cascaded upsampling approaches in continuous diffusion models (Dhariwal & Nichol, 2021; Ho et al., 2022), we experiment with using a cascaded super-resolution stage in the discrete latent space, and observe significant improvements compared to single resolution modeling (Table 2). Specifically, we train an upsampling discrete denoising model to model the sequence of high-resolution visual tokens, conditioned on the low-resolution sequence. The upsampling model is trained with the objective of (2), using a cosine masking schedule. We refer to Appendix C for further details on the upsampling stage.

**Implementation Details**  The input images are discretized into a set of 10-bit integers using a VQGAN encoder (Esser et al., 2021b), *e.g.* a 512×512 image is represented as a grid of 32×32 integer indices over a codebook of 1024 elements. For class-conditional generation, we prepend a class token to the sequence of visual tokens. The denoiser $\tilde{p}_\theta$ and corrector $p_\phi$ are transformer models in which, following MaskGIT (Chang et al., 2022) and Token-Critic (Lezama et al., 2022), the denoiser has 24 layers and 16 heads. Importantly, we used the same denoiser $\tilde{p}_\theta$ for MaskGIT, Token-Critic and the proposed DPC, which was trained using maximum likelihood of the one-step model in (2). Thus, the comparison between these methods is restricted to the sampling scheme. The corrector is a 20-layer, 12-head transformer. The denoiser was trained for 600 epochs and the corrector for 300 epochs on 32 TPUs v4 with batch size 256. The VQGAN encoder was trained at 256×256 resolution on the same datasets. The upsampling stage is a 16layer transformer and was trained for 300 epochs to denoise the encodings of 512×512 images conditioned on the encoding of ground truth 256×256 images. For DPC, unless otherwise noted, we use $T = 18$ diffusion steps, $C = 5$ and $\tau_1 = 4, \tau_2 = 5$. We control the sampling temperature of $p_\phi$ with the scale parameter $\beta$ of the Gumbel noise, which is set to $\beta = 0.6$ for DPC-full and $\beta = 0.5$ for DPC-light.

## 4.2  MAIN RESULTS

**Quantitative Evaluation**  Table 1 presents the quantitative comparison for the class-conditional generation on ImageNet. To compare the base modeling capacity of each method, no external pretrained classifiers or upsampling stages are used during training or sampling. As shown, the proposed DPC yields the best IS and FID scores on both 256×256 and 512×512 resolutions. DPC-light(5) achieves a good balance between the fidelity and the number of function evaluations (correction + time-transitions). For example, DPC-light(5) outperforms the previous best discrete diffusion VQ-Diffusion model by a large margin in FID while using fewer diffusion steps. It improves the

| Model | Type | NFE | ImageNet 256×256 | | | | ImageNet 512×512 | | | |
| --- | --- | --- | --- | --- | --- | --- | --- | --- | --- | --- |
| | | | FID ↓ | IS ↑ | Prec ↑ | Rec ↑ | FID ↓ | IS ↑ | Prec ↑ | Rec ↑ |
| BiGGAN | *GAN* | 1 | 6.95 | 202.6 | **0.86** | 0.24 | 8.43 | 177.9 | **0.85** | 0.25 |
| ADM | *cont.* | 250 | 10.94 | 101.0 | 0.69 | **0.63** | 23.24 | 58.06 | 0.73 | **0.60** |
| CDM | *cont.* | 250 | 4.88 | 158.7 | – | – | – | – | – | – |
| MaskGIT | *transf.* | 18 | 6.56 | 203.6 | 0.79 | 0.48 | 8.48 | 167.1 | 0.78 | 0.46 |
| MaskGIT ($T$=66) | *transf.* | 66 | 5.09 | 198.7 | 0.79 | 0.52 | 7.10 | 166.6 | 0.77 | 0.50 |
| Token-Critic | *transf.* | 36 | 4.69 | 174.5 | 0.76 | 0.53 | 6.80 | 182.1 | 0.73 | 0.50 |
| VQ-Diffusion | *discrete* | 100 | 11.9 | – | – | – | – | – | – | – |
| DPC-full(5) | *discrete* | 180 | **4.45** | 244.8 | 0.78 | 0.52 | **6.06** | 218.9 | 0.80 | 0.47 |
| DPC-light(5) | *discrete* | 66 | 4.8 | **249.0** | 0.80 | 0.50 | 6.09 | **228.1** | 0.81 | 0.46 |

Table 1: Bare model comparison on class-conditional generation on ImageNet (models do not use external classifiers or upsampling during sampling or training). We list the model type where "cont." is short for continuous diffusion, "discrete" for discrete diffusion, and "transf." for transformer.

| Model | NFE | FID ↓ | IS ↑ | Prec ↑ | Rec ↑ |
|-------|-----|-------|------|--------|-------|
| SyleGAN-XL | 1 | 3.58 | 219.8 | 0.73 | 0.43 |
| ADM+G+U | 250×3 | 3.85 | 221.7 | **0.84** | 0.53 |
| MaskGIT+U | 18+6 | 5.10 | 221.8 | 0.80 | 0.51 |
| MaskGIT+R+U | 18×5 | 4.26 | 326.9 | 0.81 | 0.50 |
| DPC-light(5)+U | 66+6 | 3.62 | 249.4 | 0.77 | **0.56** |
| DPC-light(5)+R+U | 66×5+6 | **3.54** | **350.2** | 0.78 | 0.55 |

Table 2: Performance of methods that leverage an up-sampling stage (U) and/or a pre-trained classifier for guidance (G) or rejection (R), on ImageNet 512x512.

| Model | quality | diversity |
|-------|---------|-----------|
| StyleGAN-XL | 77.6% ±0.42 | 54.4% ±0.50 |
| ADM+G+U | 67.6% ±0.42 | 53.2% ±0.50 |
| MaskGIT | 68.8% ±0.46 | 54.0% ±0.50 |
| Token-Critic | 68.0% ±0.46 | 48.4% ±0.50 |

Table 3: Proportion of times our DPC-light(5) model is preferred over other state-of-the-art class-conditional generation models in the user studies conducted on the ImageNet (512×512) benchmark.

quality of non-autoregressive transformers MaskGIT and Token-Critic with a reasonable increase in NFE (36) which is still less than that of the continuous diffusion models (250). We show DPC's performance for the unconditional generation on the Places2 dataset in Table 4.

In Table 2, we examine the comparison when the class-conditional generation is aided by pre-trained classifiers used as gradient guidance (G) or rejection samplers (R), or when leveraging an upsampling stage (U) for cascaded generation. Using external classifiers improves the generation quality but makes it difficult to compare models given the distinct types of classifier architectures being used such as DeiT (Touvron et al., 2021) in SyleGAN-XL, UNet with CLIP attention (Radford et al., 2021) in ADM, or ResNet (He et al., 2016) in MaskGIT. Nevertheless, we present results for this setup by using a rejection sampling scheme based on a ResNet-50 classifier pretrained on ImageNet, with a specified acceptance rate of 25%. We apply the rejection sampling in samples generated at 256×256 resolution and then apply 6 upsampling decoding iterations to obtain the final 512×512 samples. For reference, ADM+G+U (Dhariwal & Nichol, 2021) uses 128×128 generation followed by upsampling to 512×512. DPC achieves the best FID and Inception Score under this setup among the models we consider, and is highly competitive when using only generation and upsampling.

**User Preference Study**    To further understand the perceptual differences between the compared models, we conduct two user preference studies on Amazon Mechanical Turk to verify the visual quality and diversity of our model's samples. We compare our DPC-light(5), without classifier rejection or upsampling, to StyleGAN-XL, ADM with guidance and upsampling, and MaskGIT and Token-Critic without classifier rejection. The compared images are generated using the public models of StyleGAN-XL, ADM, and MaskGIT, obtained from their official websites.

For the *quality* test, we present two randomly sampled images of the same class side-by-side (one from DPC-light and one from the compared model), and ask the graders to select which one looks more realistic. For the *diversity* test, we show graders two sets of six randomly sampled images each for four different classes, and ask them to select which set is more diverse looking. The quality and diversity tests are carried out on each of the 1,000 ImageNet classes, with 14 different graders, for a total of 14,000 comparisons. The comparison to each method is organized into 250 task groups.

As shown in Table 3, DPC is preferred for *quality* more times than all the compared methods. It is noteworthy that the compared models in Table 3 represent the best published class-conditional models on the ImageNet benchmark, including GAN, diffusion, and transformer models. The differences in the quality test are statistically significant at the $p$-value level of 0.05. It also appears that DPC shows preferred *diversity* compared to StyleGAN-XL and ADM+G+U.

**Qualitative Evaluation**    In Figure 1, we show random samples from our ImageNet 512×512 DPC-light(5) model, and from the publicly available implementations of StyleGAN-XL and ADM with classifier guidance and upsampling. These are the same models used in the user studies. We refer to Appendix D and the supplementary material for more comprehensive qualitative comparisons.

## 5    RELATED WORK

**Diffusion Models**    The most prominent type of diffusion model that operates entirely in continuous space is the Gaussian diffusion model (Sohl-Dickstein et al., 2015; Song & Ermon, 2019; Ho et al., 2020; Song et al., 2021; Kingma et al., 2021; Tzen & Raginsky, 2019; Kadkhodaie & Simoncelli,

| Model | NFE | FID ↓ | IS ↑ | Pre | Rec |
|---|---|---|---|---|---|
| MaskGIT | 18 | 31.3 | 8.6 | **0.65** | 0.42 |
| MaskGIT | 66 | 23.3 | 9.9 | 0.62 | 0.47 |
| DPC-full(1) | 36 | 20.3 | 9.6 | 0.62 | 0.47 |
| DPC-full(5) | 180 | 16.3 | **12.1** | 0.59 | **0.50** |
| DPC-light(5) | 66 | **16.2** | 11.5 | 0.55 | **0.50** |

Table 4: Performance on unconditional generation in the Places2 validation dataset at 512×512. We include MaskGIT's performance for reference.

| | DPC-full ($\beta$=0.6) | | | DPC-light ($\beta$=0.5) | | |
|---|---|---|---|---|---|---|
| $C$ | NFE | FID↓ | IS↑ | NFE | FID↓ | IS↑ |
| 1 | 36 | 23.87 | 101.0 | 26 | 15.69 | 128.9 |
| 2 | 72 | 10.59 | 162.9 | 30 | 10.77 | 161.3 |
| 3 | 108 | 7.25 | 194.5 | 38 | 8.13 | 187.6 |
| 4 | 144 | 6.23 | 211.3 | 50 | 7.13 | 202.2 |
| 5 | 180 | 6.06 | 218.9 | 66 | 6.09 | 228.1 |
| 6 | 216 | 6.20 | 222.1 | 86 | 5.99 | 233.0 |
| 7 | 252 | 6.53 | 221.1 | 108 | 5.86 | 245.1 |

Table 5: ImageNet 512×512 performance when varying $C$ (corrections per timestep).

2021), which is formulated as a forward and a learned reverse process that are both parameterized as conditional Gaussian distributions. These models have attained high quality generation results in image and audio tasks (Dhariwal & Nichol, 2021; Ho et al., 2022; Saharia et al., 2021b;a; Nichol et al., 2021; Ramesh et al., 2022; Chen et al., 2020b; Kong et al., 2021b; Lugmayr et al., 2022).

**Discrete Diffusion** Recently, D3PM (Austin et al., 2021) and OA-ARDM (Hoogeboom et al., 2022) applied discrete diffusion (Sohl-Dickstein et al., 2015; Hoogeboom et al., 2021) to image modeling, focusing on density estimation and compression of raw image pixels. Campbell et al. (2022) introduce a framework to model discrete diffusion in continuous time, and demonstrate that a combination of the denoising model and forward noising process is a valid corrector. In contrast to Campbell et al. (2022), our learned corrector can directly target the most unlikely tokens, achieving efficient sampling with one order of magnitude fewer steps. While these methods limit their analysis to low-resolution images (*e.g.*, 32×32), VQ-Diffusion (Gu et al., 2022) is the state-of-the-art discrete diffusion model for ImageNet image synthesis, also employing a mask-and-replace diffusion strategy to predict missing VQ-GAN tokens iteratively. Different from our model, VQ-Diffusion may still suffer from the compounding decoding error discussed in this paper. Our results show that our model outperforms VQ-Diffusion in both quality and efficiency on the ImageNet benchmark, while further scaling discrete diffusion models from the 256×256 to the 512×512 resolution.

**Non-autoregressive transformers for image synthesis** While early works (van den Oord et al., 2016; Salimans et al., 2017; Parmar et al., 2018; Chen et al., 2020a) modeled images directly in pixel space, various recent works (Ramesh et al., 2021; Esser et al., 2021b;a; Ding et al., 2022) used autoregressive transformers over the discrete latent space provided by VQVAE or VQGAN. Closest to our work is a class of non-autoregressive transformers (Chang et al., 2022; Zhang et al., 2021; Lezama et al., 2022; Kong et al., 2021a) which have recently demonstrated improved quality and efficiency over conventional autoregressive transformer models like Esser et al. (2021b). There are two major differences to these non-autoregressive transformers. First, we propose new techniques to perform multi-step correction steps which, as shown in our experiments, are essential for achieving a desirable quality-vs-efficiency trade-off. Second, we show that, in the ideal case, the proposed method can correct the train/test mismatch in the distributions of intermediate states. Finally, we further demonstrate the applicability of cascaded upsampling in the discrete latent space.

# 6 CONCLUSION

Parallel sampling from generative transformers dramatically improves efficiency compared to full autoregressive sampling. However, from the perspective of discrete diffusion models, it exacerbates deviations from the ideal sampling distribution that accumulate in the marginal distributions of intermediate diffusion states. To mitigate this compounding decoding error, we proposed DPC, a new discrete diffusion model based on a learned MCMC corrector kernel that refines the samples of these intermediate states. Empirically, we demonstrated that the repeated application of the learned corrector improves the samples of a non-autoregressive transformer, as measured on the standard ImageNet class-conditional generation task. User preference studies showed that DPC is competitive with state-of-the-art generative vision transformers, GANs, and continuous diffusion models.

ACKNOWLEDGEMENTS

We would like to thank Ming-Hsuan Yang and Douglas Eck for helpful comments during early stages of this work. We also thank the anonymous reviewers for their insightful comments and constructive feedback that helped to improve this paper.

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

## A    LEARNING THE MASK MODEL

Let $\{l^i\}_{i=1...N}$ be a set of N logits, given by the output of a neural network taking $\hat{\boldsymbol{x}}_0, t$ as input, i.e. $l^i = l^i_\phi(\hat{\boldsymbol{x}}_0, t)$. We perturb these logits by adding i.i.d. Gumbel noise, i.e. $\tilde{l}^i = l^i + G^i$ with $G^i \sim \text{Gumbel}(0, 1)$, and we rank-order these perturbed logits to get a ranking $\boldsymbol{c}$. Then the *Plackett-Luce model* (Plackett, 1975) states that the probability of obtaining an ordering $\boldsymbol{c}$ is given by

$$p(\boldsymbol{c}|\hat{\boldsymbol{x}}_0) = \prod_{i=1}^{N} \frac{\exp(l^{c^i})}{\sum_{j=i}^{N} \exp(l^{c^j})}. \tag{11}$$

If we only care about the top-k ranking $\boldsymbol{c}_k = \{c^1, \ldots, c^k\}$, then the probability of that partial ranking can be obtained as

$$p(\boldsymbol{c}_k|\hat{\boldsymbol{x}}_0) = \prod_{i=1}^{k} \frac{\exp(l^{c^i_k})}{\sum_{j=i}^{k} \exp(l^{c^j_k}) + R}, \tag{12}$$

with $R = \sum_{j=(k+1)}^{N} \exp(l^{c^j})$.

We define our mask $\boldsymbol{m}_t$ to be a vector with one boolean element $m^i_t$ for each element $\tilde{l}^i$. If $\tilde{l}^i$ is among the $k_t$ largest elements we set $m^i_t = 1$, and otherwise we set $m^i_t = 0$. Using numpy notation, we now have `np.sort(`$\boldsymbol{c}_k$`) = np.nonzero(`$\boldsymbol{m}_t$`)`. For sampling a mask $\boldsymbol{m}_t$ we don't care about the exact order $\boldsymbol{c}_k$, but only about the elements in this top-k ordering. The probability of sampling a mask $\boldsymbol{m}_t$ can thus be found by summing over all the partial rankings $\boldsymbol{c}_k$ that are consistent with this mask:

$$p(\boldsymbol{m}_t|\hat{\boldsymbol{x}}_0) = \sum_{\boldsymbol{c}_k(\boldsymbol{m}_t)} \prod_{i=1}^{k} \frac{\exp(l^{c^i_k})}{\sum_{j=i}^{k} \exp(l^{c^j_k}) + R} \tag{13}$$

$$= \exp\left(\sum_{i:\boldsymbol{m}^i_t=1}^{k} l^i\right) \sum_{\boldsymbol{c}_k(\boldsymbol{m}_t)} \prod_{i=1}^{k} \frac{1}{\sum_{j=i}^{k} \exp(l^{c^j_k}) + R},$$

$$\text{with } R = \sum_{l:\boldsymbol{m}^l_t=0} \exp[l^l_\phi(\hat{\boldsymbol{x}}_0, t)]. \tag{14}$$

Evaluating the mask likelihood in (13) requires summing over all partial orderings $\boldsymbol{c}_k$ consistent with the mask $\boldsymbol{m}_t$, which is computationally expensive. Next we discuss two alternative formulations that are more computationally efficient.

Under the Plackett-Luce model, the logits $l^i, l^j$ encode the relative log-odds of $i$ ending higher than $j$ in the sampled ranking $\boldsymbol{c}$. That is $p(\tilde{l}^i > \tilde{l}^j) = \sigma(l^i - l^j)$. (When applied to *pairs* of elements like this, the Plackett-Luce model is also known as the *Bradley-Terry model* (Bradley & Terry, 1952; Hunter, 2004; Huang et al., 2006).) Thus, one alternative is to consider the relative scores of masked and unmasked elements, which yields the following objective:

$$\mathcal{L}_{pairwise} = \mathop{\mathbb{E}}_{\substack{(\boldsymbol{x}_t, \boldsymbol{m}_t) \sim q(\boldsymbol{x}_t, \boldsymbol{m}_t) \\ \hat{\boldsymbol{x}}_0 \sim p_\theta(\hat{\boldsymbol{x}}_0|\boldsymbol{x}_t)}} \sum_{i:m^i_t=1}^{k} \sum_{j:m^j_t=0}^{N-k} -\log\left(\sigma\left(l^i_\phi(\hat{\boldsymbol{x}}_0, t) - l^j_\phi(\hat{\boldsymbol{x}}_0, t)\right)\right). \tag{15}$$

Another, simpler loss function which we found to work well in practice, is binary cross entropy independently applied to each mask element using $p_\phi(m^i_t = 1|\hat{\boldsymbol{x}}_0, t) \approx \sigma(l^i_\phi(\hat{\boldsymbol{x}}_0, t))$, where $\sigma$ is the sigmoid function. The reasoning here is that, if the cutoff between ending in the top-k and ending outside of it is represented by $\bar{l}$, we may reasonably approximate the probability of ending in the top-k as $p(i \in \boldsymbol{c}_k) \approx \sigma(l^i - \bar{l})$. Furthermore, since the Plackett-Luce distribution is invariant to shifts in the logits $l^i$, the absolute level of $\bar{l}$ can be chosen at will. We therefore choose $\bar{l} = 0$, resulting in $p_\phi(m^i_t = 1|\hat{\boldsymbol{x}}_0, t) \approx \sigma(l^i_\phi(\hat{\boldsymbol{x}}_0, t))$, which we fit against the marginal distribution of observed mask

| Model | FID ↓ | IS ↑ | Prec ↑ | Rec ↑ |
|---|---|---|---|---|
| *factorized* | **6.41** | 209.5 | **0.78** | **0.45** |
| *pairwise* | 10.89 | **231.6** | 0.77 | 0.38 |

Table 6: Performance comparison between corrector models trained with the factorized loss (16) and the pairwise loss (15), for the same number of training steps (200 epochs), on ImageNet 512x512. In both cases we use the DPC-full(5) sampling schedule.

elements $q(m_t^i)$ using maximum likelihood:

$$\mathcal{L}_{factorized} = \mathop{\mathbb{E}}_{\substack{(\boldsymbol{x}_t, \boldsymbol{m}_t) \sim q(\boldsymbol{x}_t, \boldsymbol{m}_t) \\ \hat{\boldsymbol{x}}_0 \sim p_\theta(\hat{\boldsymbol{x}}_0 | \boldsymbol{x}_t)}} -\sum_{i=1}^{k} m_t^i \cdot \log\left(\sigma\left(l_\phi^i(\hat{\boldsymbol{x}}_0, t)\right)\right) - (1 - m_t^i) \cdot \log\left(1 - \sigma\left(l_\phi^i(\hat{\boldsymbol{x}}_0, t)\right)\right).$$

(16)

Table 6 shows a quantitative comparison of the corrector models trained with $\mathcal{L}_{pairwise}$ and $\mathcal{L}_{factorized}$. We train both models for the same number of training steps (200 epochs), and sweep sampling hyperparameter ($\beta$) for better FID. We found the factorized loss to be more efficient and we use this loss in all the experiments reported in the main manuscript.

## B    ABLATION STUDIES

In Figure 2 we show resulting FID and Inception Score when varying the parameter for the sampling temperature $\beta$. Our observation is that the resulting sampling quality, as measured by FID and Inception Score, is highly sensitive to the value of $\beta$, and that the optimal $\beta$ value also depends on the number of correction steps $C$ and the refinement schedule $c(t)$.

In Figure 3, we present an ablation study on the shape of the trapezoidal function $c(t)$ for the DPC-light model. We vary the starting point of the top side of the trapezoid, $\tau_1$, and its width $\tau_2 - \tau_1$. We observe slightly better metrics by concentrating the trapezoid around the first steps of the decoding process.

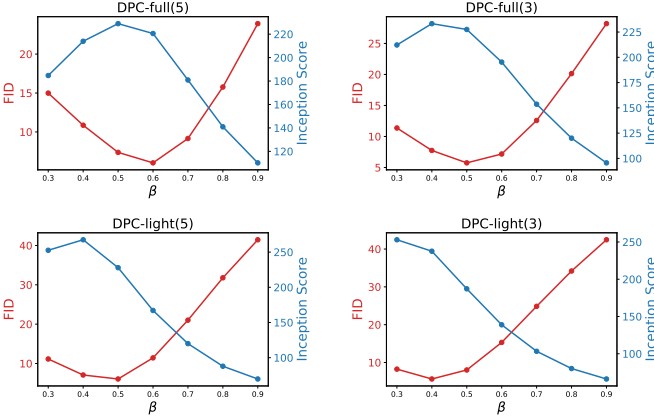

Figure 2: Ablation study for sampling temperature parameter $\beta$. We show the effect of $\beta$ on FID and Inception Score for models DPC-full(5), DPC-full(3), DPC-light(5) and DPC-light(3). We notice a high sensitivity to $\beta$ and a strong correlation between the effect on FID and Inception Score.

## C    UPSAMPLING IN DISCRETE SPACE

We perform upsampling in the discrete latent space by learning to generate the sequence of visual tokens corresponding to a high-resolution image $\boldsymbol{x}_0^H$, given the sequence of visual tokens from a low-resolution downsampled version $\boldsymbol{x}_0^L$. For the experiments in the paper we use the 512×512 and 256×256 resolutions, and use the same VQ-encoder for both resolutions, yielding sequences of

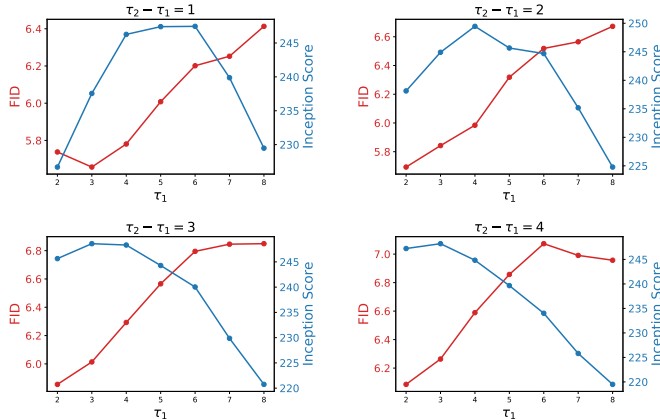

Figure 3: Ablation study for refinement schedule parameter $\tau_1$. We show the effect of varying the location of the top left vertix of the trapezoid $\tau_1$ and its width $\tau_2 - \tau_1$ on FID and Inception Score, for DPC-light(5). We observe that concentrating more refinement steps towards the begining of the decoding process is beneficial for FID, and around the $t = 4$ for Inception Score. In all experiments the number of diffusion steps is $T = 18$.

| Masking Rate | FID $\downarrow$ | IS $\uparrow$ | Prec $\uparrow$ | Rec $\uparrow$ |
|---|---|---|---|---|
| 0% | **5.19** | **174.5** | 0.78 | 0.50 |
| 5% | 5.22 | 173.5 | 0.78 | 0.50 |
| 10% | 5.28 | 172.0 | 0.78 | 0.50 |

Table 7: Effect of random perturbation applied to the low-resolution sequence when training the upsampling stage. The perturbation consists in masking a portion of the tokens and replacing them with the predictions of the low-resolution denoising model. We use the one-step denoising model (without corrector) for this experiment.

length 1024 (32×32) and 256 (16×16), respectively. The upsampling model $p_u$ is trained with the one-step denoising objective:

$$\mathcal{L}_{upsampling} = \mathop{\mathbb{E}}_{(\boldsymbol{x}_0^L, \boldsymbol{x}_0^H) \sim q} - \log \prod_i p_u \left( x_0^H(i) | \boldsymbol{x}_t^H, \boldsymbol{x}_0^L, \boldsymbol{m}_t \right), \tag{17}$$

where the mask rate in $\boldsymbol{m}_t$ follows a cosine scheduling function as for the denoiser model $\tilde{p}_\theta$. We use a transformer to model the high resolution sequence, with cross-attention to the low-resolution sequence in every layer and independent positional encodings for each resolution.

During inference, we first generate an image (sequence of visual tokens $\hat{\boldsymbol{x}}_0^L$) in 256×256 resolution. We then perform 6 decoding steps with the upsampling model $p_u$, following the cosine scheduling function. For simplicity, when using DPC with upsampling, we only apply corrector steps in the low resolution, and use the prediction confidence in $p_u$ to keep/reject tokens (similarly to MaskGIT (Chang et al., 2022)) in each upsampling decoding iteration.

Following observations in cascaded continuous diffusion, where the low resolution images are augmented with random perturbations to reduce exposure bias (Ho et al., 2022), we experimented with randomly perturbing the low-resolution sequence. To randomly perturb the low-resolution sequence we randomly mask a subset of tokens and ran the partially masked sequence through the low-resolution denoiser $\tilde{p}_\theta$. Table 7 shows the results obtained for different strengths of perturbation (masking rate). These results suggest that in the discrete case, low-resolution augmentation does not improve the upsampling stage. Thus, for simplicity we did not use any augmentation for the experiments in Section 4.

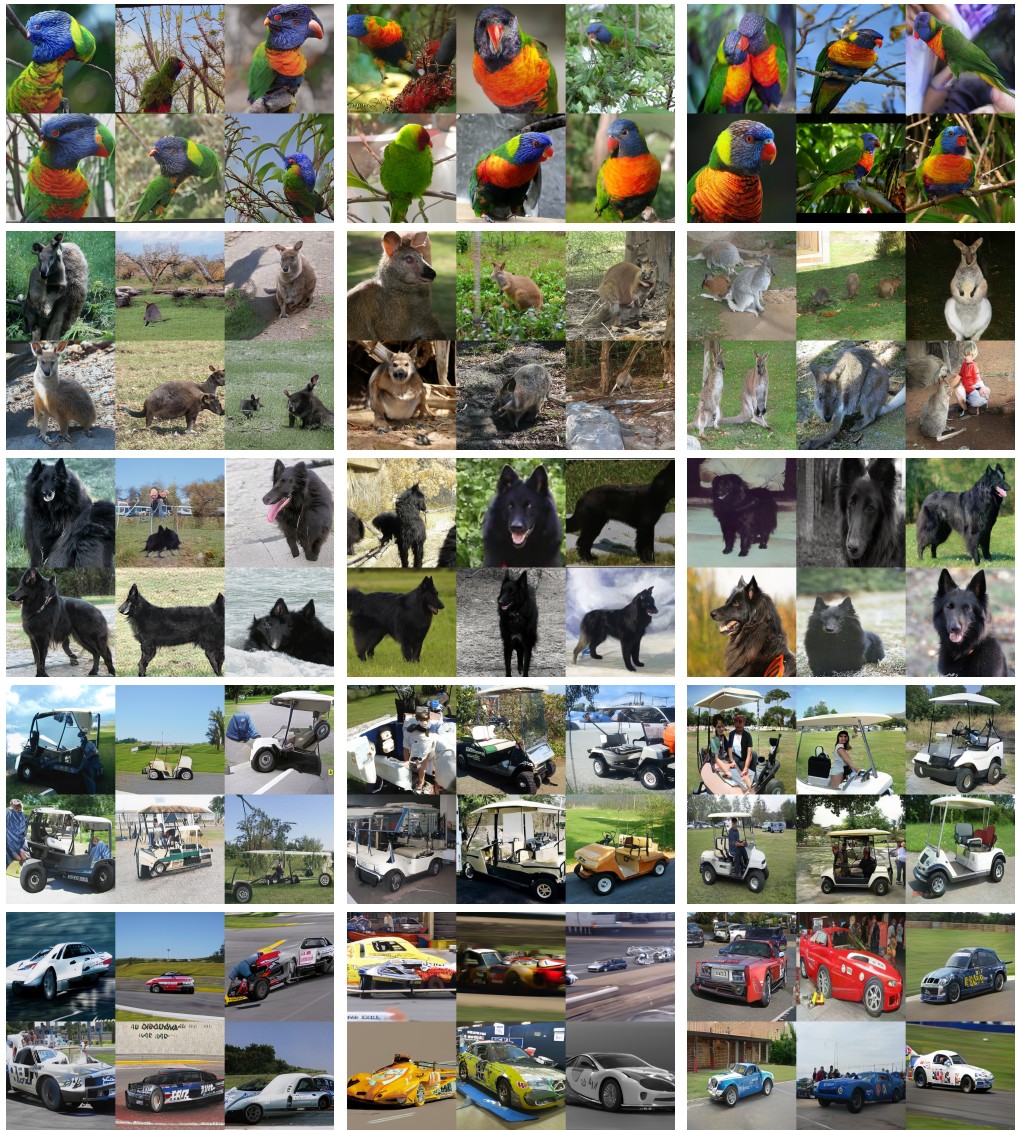

Figure 4: Random samples from ImageNet 512×512 class-conditional generation for selected classes: 'lorikeet', 'wallaby', 'flat-coated retriever', 'groenendael', 'golfcart' and 'racing car'. **Left:** StyleGAN-XL (NFE = 1). **Center:** ADM + Classifier Guidance + Upsampling (NFE = 250 × 3). **Right:** DPC-light(5) (NFE = 66).

## D    FURTHER QUALITATIVE RESULTS

**ImageNet**   We present further 512x512 ImageNet class-conditional samples in Figure 4, comparing the proposed DPC-light(5) to StyleGAN-XL (Sauer et al., 2022) and ADM with Guidance and Upsampling  (Dhariwal & Nichol, 2021). In Figure 5 we further show random samples of random classes for each of the three methods. We provide further qualitative results in the Supplementary Material.

**Places2**   In Figure 6 we show samples from the Places2 dataset (Zhou et al., 2017), by the proposed DPC-light(5) (FID 16.2) and MaskGIT (FID 26.3), as well as original images from the dataset for reference.

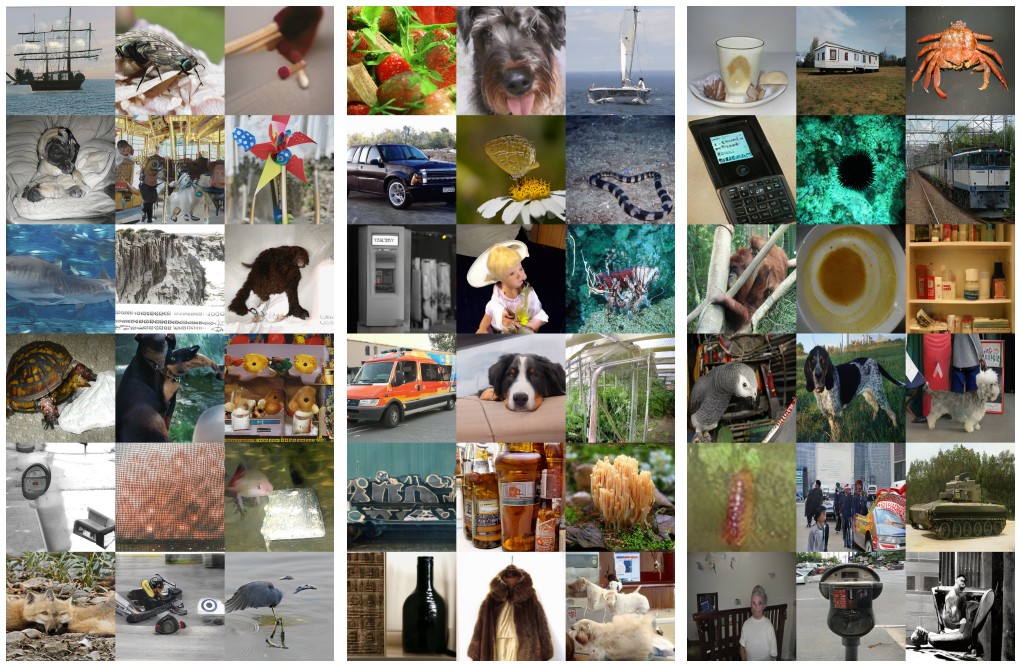

Figure 5: Random samples from random classes from ImageNet 512×512 class-conditional generation for selected classes. **Left:** StyleGAN-XL (NFE = 1). **Center:** ADM + Classifier Guidance + Upsampling (NFE = 250 × 3). **Right:** DPC-light(5) (NFE = 66).

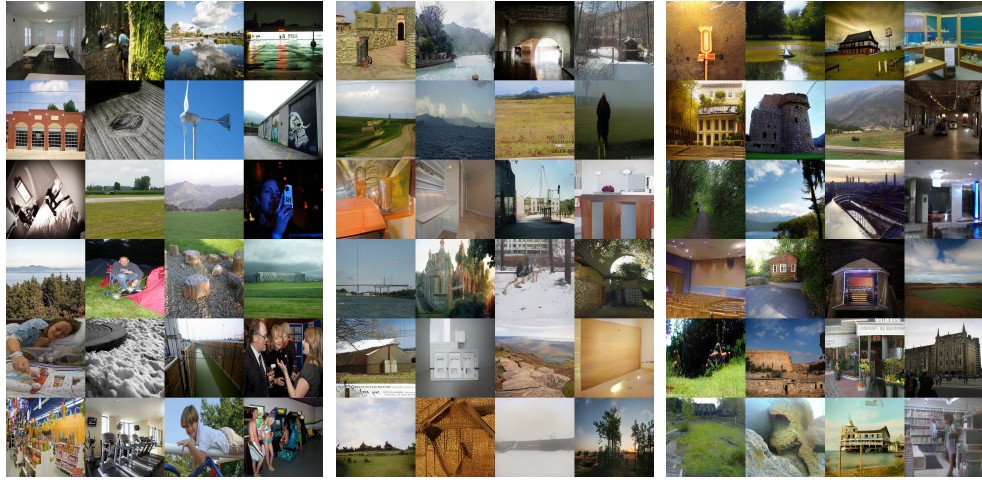

Figure 6: Unconditional image generation in the Places2 dataset (Zhou et al., 2017) at 512 × 512 resolution. Random samples from **left:** Ground Truth, **center:** MaskGIT (FID 23.3, NFE = 66), **right:** DPC-light(5) (FID 16.2, NFE = 66).

## E  FURTHER IMPLEMENTATION DETAILS

All the transformers used in this work have embedding dimension 768 and hidden dimension 3,072, learnable positional embedding (Devlin et al., 2019), LayerNorm (Ba et al., 2016), and truncated normal initialization (stddev= 0.02). The following training hyperparameters were used for both transformers: dropout rate 0.1, Adam optimizer (Kingma & Ba, 2014) with $\beta_1 = 0.9$ and $\beta_2 = 0.96$. We used *RandomResizeAndCrop* for data augmentation and the denoiser is trained with label smoothing set to 0.1.

## F    USER PREFERENCE STUDY DETAILS

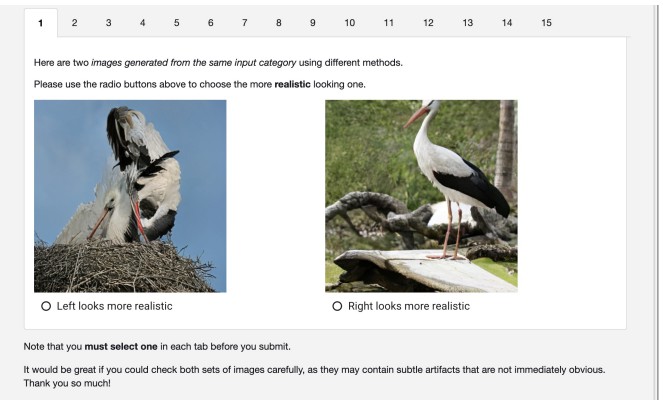

Figure 7: Screenshot for the *quality* user preference study.

In Figure 7 we present a screenshot for the *quality* user study. Users were presented with two generated images of the same class, one sampled from our method and one from the compared method, and were shown the following text prompt:

> Here are two images generated from the same input category using different methods.
>
> Please use the radio buttons above to choose the more realistic looking one.
>
> It would be great if you could check both sets of images carefully, as they may contain subtle artifacts that are not immediately obvious. Thank you so much!

In Figure 8 we show a screenshot for the *diversity* user preference study. Users were presented with two grids of random samples, one for our method and one for the compared method. The grid contained 6 images for each of 4 classes. The users were shown the following text prompt:

> In each tab, we have two sets of images generated from the same input categories using different methods.
>
> Please use the radio buttons above to choose the image set with more diverse content, lighting and background.
>
> Note that you must select one in each tab before you submit.
>
> It would be great if you could check both sets of images carefully, as they may contain subtle differences that are not immediately obvious. Thank you so much!

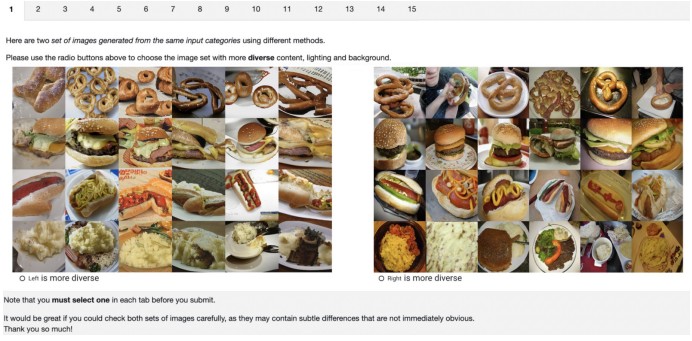

Figure 8: Screenshot for the *diversity* user preference study.

