# OpenReview forum: "Discrete Predictor-Corrector Diffusion Models for Image Synthesis"
_ICLR.cc/2023/Conference — ICLR 2023 poster_

### Official Review · Reviewer_mMR5 · 2022-10-20

**Confidence:** 3
**Correctness:** 3
**Technical Novelty And Significance:** 4
**Empirical Novelty And Significance:** 4
**Recommendation:** 8

**Clarity, Quality, Novelty And Reproducibility:**

The paper is clearly written and nice to read. Related work is clearly described.

No code is provided, which limits the usefulness and reproducibility of the work.

4.1 – please explain how the class conditioning is done.


**Details Of Ethics Concerns:**

The paper provides new tools for creating synthetic images that look like real photos, which could be used in both bad and good ways.

**Strength And Weaknesses:**

The results are impressive. The sampling method is elegant and some reasonable argument is made about why it works.

I would like to see more discussion of why this new learned corrector works well. The only explanation really offered by the authors in 3.1 is that it approximately satisfies detailed balance for $q(x_t)$, but many different correctors would approximately satisfy detailed balance for this distribution, and some will have better properties (mixing time, accuracy of the stationary distribution) than others. Campbell et al. 2022 http://arxiv.org/abs/2205.14987 introduce a simple corrector for discrete-state-space diffusion models, which requires no extra training. Compared to the present learned corrector, Campbell’s corrector seems like a closer analogue to the Langevin corrector described by Song et al.. Campbell et al. also justify their corrector in terms of detailed balance. It does not have the flexibility to selectively re-mask those pixels that do not ‘match’ other pixels (which I expect the learned corrector to do), and intuitively I expect the simple corrector to have lower-entropy transitions and longer mixing time than the learned corrector. It would be nice to see an empirical comparison.

A theoretical question - does small KL divergence in (10) imply small deviation from the ideal stationary distribution in (6)? It’s possible for two transition  matrices to be arbitrarily close to each other as measured using e.g. Lp norm, while having very different stationary distributions.

Lugmayr et al. 2022  https://arxiv.org/abs/2201.09865 also do (continuous state-space, conditional) image generation by interleaving forward and reverse diffusion steps.  How does it relate to the present work?


**Summary Of The Paper:**

The paper presents a new sampling method that improves the quality of generated images from discrete diffusion models.  In the new method, denoising steps are interleaved with ‘corrector’ steps that mitigate the problem of compounding decoder error.

**Summary Of The Review:**

The paper represents a useful advance in improving the cost/quality tradeoff when sampling images using discrete diffusion models. It introduces a technique that could be more broadly useful.

---

> ### Author Response · Authors · 2022-11-17
> **Response to Reviewer mMR5**
>
> Thank you for your insightful review. We reply to your remarks below.
>
> **Why the corrector works well. Comparison to Campbell et al.:**
> Thank you very much for pointing out this highly relevant reference, which we will add to the manuscript.
>
> In the absorbing state discrete diffusion, the reverse or generative iterations consist in refinement by rejection and resampling. We believe that the critical aspect introduced by the learned corrector is an improved rejection by learning to target unlikely elements.  In the case of the absorbing state diffusion, the corrector
> proposed in Campbell et al. would be derived from the combination of the
> random masking transition matrix and the transition matrix given by
> the reverse model, in our case the denoiser. This would be adding
> probability mass to the transition between any state and the masked
> state.  Since under our denoiser model this probability is 0, for such corrector all
> tokens would have an equal probability of transitioning to the mask
> state.  One can expect this random rejection to be slower to converge than the targeted rejection of our learned corrector. Although Campbell et al.'s experiments are for the uniform diffusion, the much higher number of model evaluations (1000 vs 66 for our light model), suggests our corrector converges indeed faster. We leave a rigorous experimental comparison under the same setting for future work.
>
> **Small KL divergence:**
> We can say that if the KL divergence in (10) is 0, then the stationary distribution is ideal. One could indeed construct adversarial cases where the KL between the transition distributions is very small, but the KL between stationary distributions is large. We hypothesize these
> cases do not naturally occur often, since otherwise the method would not work. Moreover, even in such adversarial cases we would generally still have that the KL between stationary distributions will decrease if the KL between transition distributions further decreases, i.e. the training signal generally still points in the right direction.
>
> **Relation to Lugmayr et al. 2022:**
> Thank you for this reference. Using the forward process to perform correction as in Lugamyr et al. is very interesting. In our case, simply swapping out $p_\phi(x_t|\hat{x}_0)$ for $q(x\_t|\hat{x}\_0)$ amounts to applying random masking to the sample $\hat{x}\_0$, instead of the mask sampled by the corrector. This did not work in our experiments, failing to produce meaningful images. One intuitive explanation is that in this case, the subsequent iterates $x\_t,x\_{t-1}$ will be nearly independent early on in sampling, and the model will be unable to improve the samples with just local iterative refinement.
>
>
>
> **Clarity, Novelty and Reproducibility:**
> The class conditioning is obtained by prepending a class token to the input sequence. We will clarify this in the manuscript.
>
> We plan to make the code publicly available if accepted.

---

> > ### Comment · Reviewer_mMR5 · 2022-12-13
> > **Thank you for the response.**
> >
> > I'm glad to hear that you plan to make the code publicly available.

---

### Official Review · Reviewer_bius · 2022-10-25

**Confidence:** 4
**Correctness:** 3
**Technical Novelty And Significance:** 3
**Empirical Novelty And Significance:** 3
**Recommendation:** 8

**Clarity, Quality, Novelty And Reproducibility:**

Clarity: 7/10,
Quality: 7/10,
Novelty: 6/10,
Reproducibility 7/10.

**Strength And Weaknesses:**

Strength:
(1) The paper is well-written and easy to follow.
(2) The discrete predictor-corrector algorithm makes sense.
(3) The experimental results are stable, showing this model is effective. The results are evaluated on some widely used metrics, and the user study is reasonable.

Weaknesses:
(1) The diffusion model is computationally expensive, so it would be better to analyze the computation cost or time consumption for training and inference.
(2) The paper focuses on ImageNet and Places2 datasets. These datasets have less space to improve compared with complex datasets. So it would be better to include more datasets (e.g., LSUN, CelebAHQ). It is also worth trying to contain many objects in one image, although it is difficult to deal with.

**Summary Of The Paper:**

This paper proposes the Discrete Predictor-Corrector diffusion model (DPC). This model can synthesis better images, which is discussed both qualitatively and quantitatively. This model is evaluated on class-conditional image generation on the ImageNet dataset and unconditional generation on the Places2 dataset.

**Summary Of The Review:**

The paper is well-written and easy to follow. The experimental results are stable. While the main claim is that the DPC model improves the generated images' quality compared with previous discrete diffusion. The novelty would raise if this paper could show that the DPC model extends the ability or accelerate the computation of the diffusion model.

---

> ### Author Response · Authors · 2022-11-17
> **Response to Reviewer bius, part 1.**
>
> Thank you very much for your review.
>
> (1) We agree that the continuous diffusion model is computationally expensive. As stated in the introduction, our model offers far better efficiency than continuous diffusion models as it performs discrete diffusion on a latent space.
> For example, comparing 512x512 image generation speeds on a V-100 GPU with default settings, the continuous diffusion model ADM+G+U (Number of Function Evaluations (NFE) =750) takes approximately 45 sec/image, DPC-light (NFE=66) 4.5 sec/image and MaskGIT (NFE=18) 2 sec/image. Moreover, using 4 TPUv3 chips, and batch size 128, a compiled version of DPC-light (NFE=66) can achieve 0.16 sec/image. We summarize these results in the following table. We underline that using similar sampling time (similar NFE), the performance of DPC is still superior to that of MaskGIT (recall the corrector transformer is slightly smaller than the denoiser one, i.e. 20 vs 24 layers).
>
> | Method |NFE |sec/img on V-100 |sec/img on TPUv3 (compiled) |FID |IS |
> |----------|------|---------------------|----------------------------------|----|----|
> |ADM+G+U | 750 | 45  |  - |3.85 |221.7 |
> |MaskGIT|18|2|0.06| 7.45| 172.9|
> |MaskGIT|64|2|0.19| 7.53| 133.7|
> |DPC-light(5)| 66|4.5|0.16|6.09|228.1|
>
>
> For ImageNet-512x512, training the transformer models of the denoiser and the corrector took approximately 1 day per 100 epochs on 32 TPUv4 chips. We recall that the denoiser was trained for 600 epochs and the corrector for 300 epochs.

---

> > ### Author Response · Authors · 2022-11-17
> > **Response to Reviewer bius, part 2.**
> >
> > (2) We first point out that the Places dataset used in our experiments contains 400+ unique scenes many of which are complex and contain multiple objects (http://places2.csail.mit.edu/). Many of the LSUN categories are included in the newer and larger Places2 dataset.
> >
> > Nevertheless, to resolve the reviewer's concern we conducted an experiment to evaluate the performance of DPC on the widely used LSUN-Churches dataset. We note that due to having to train the VQ-GAN, Denoiser and Corrector one after the other in limited time, the final results are not  optimal for this setting, but the relative comparison is still relevant: as in the experiments in the main manuscript, the baseline and DPC share the same VQ-GAN and Denoiser, so the comparison focuses on the sampling scheme.
> > We present the result in the following table, where we swept the sampling temperature of each method over a grid  for the best FID, and averaged over 3 random seeds  (t-test p-value <0.002).
> > We note that applying the corrector steps in the intermediate stages improved the samples of the baseline MaskGIT when using the same diffusion steps T. Also, DPC is better than the baseline when compared at the same NFE (same computational cost of sampling).
> >
> > | Method |Diffusion steps (T) |NFEs |FID|
> > |--|--|--|--|
> > | MaskGIT |8 |8 |5.22±0.03|
> > |MaskGIT|26|26|3.26±0.03|
> > |DPC-light(3)|8|26| **3.03±0.03**|

---

### Official Review · Reviewer_spPv · 2022-10-25

**Confidence:** 3
**Correctness:** 3
**Technical Novelty And Significance:** 3
**Empirical Novelty And Significance:** 3
**Recommendation:** 8

**Clarity, Quality, Novelty And Reproducibility:**

The presentation of the idea is overall clear to me except the point I raised in the weaknesses part. The idea is novel and interesting, and the overall quality of this work is high regarding good insights and results.


**Strength And Weaknesses:**

Strengths:
- The proposed method DPC is well-motivated (to reduce the compounding decoding error introduced by the parallel decoding) and novel (by learning a new MCMC corrector kernel that serves the similar purpose of the Langevin corrector as in the continuous case).
- The insights into understanding the improving the non-autoregressive transformer methods (e.g., MaskGIT, Token-Critic) from the discrete diffusion model perspective are very interesting. It shows that the proposed method generalizes MaskGIT and Token-Critic with a better theoretical justification.
- It shows the applicability of cascaded upsampling in the discrete latent space, and observes the large improvements over single resolution modeling.
- Experiments on the challenging image datasets (ImageNet and Places2) demonstrate the effectiveness of the proposed method by comparing various state-of-the-art generative models (BigGAN, StyleGAN-XL, ADM, CDM, VQ-Diffusion, MaskGIT, and Token-Critic).
- The writing is overall clear and the paper is easy to follow.

Weaknesses:
- The corrector steps in DPC introduce more computational cost, particularly in inference. Compared to MaskGIT and Token-Critic (two similar methods as special cases of DPC without corrector steps), as shown in Table 1 and Table 3, the improvement of DPC in image quality (while it may sacrifice the diversity) comes with a larger NFE. For example, Token-Critic has 4.69 FID with 36 NFEs vs. DPC-full (5) has 4.45 FID with 180 NFEs.
- It is confusing to me how exactly the corrector steps are repeated in the shortcut time transition. I understand that the original version of DPC contains one predictor step (step 1) and one or multiple corrector steps (step 2 and step 3). Since we combine step 1 and step 2 in the shortcut time, how do we perform the predictor step and corrector steps separately?
- I think it would be better to add the background of the Predictor-Corrector sampling in the continuous case to make this work more accessible to a broader audience.


**Summary Of The Paper:**

This work extended the Predictor-Corrector sampling algorithm originally developed in the continuous diffusion models to the discrete case, which is termed discrete Predictor-Corrector (DPC). Since, in the discrete diffusion models, there is no direct counterpart to the score function, this work proposed to learn a new MCMC corrector kernel, whose limiting distribution is the true marginal distribution of the intermediate data. Besides, this work provides new insights into interpreting the parallel decoding in the non-autoregressive transformer methods as a reverse diffusion process using one-step sampling, and reveals the compounding decoding error wherein. This work focuses on applying DPC to the discrete diffusion models in the latent space of VQ-GAN. On the high-resolution image synthesis tasks, such as ImageNet (256x256 and 512x512) and Places2 (512x512), this work demonstrates that DPC performs comparable to the latest non-autoregressive methods (e.g., MaskGIT, Token-Critic) and outperforms the state-of-the-art continuous diffusion models (e.g., ADM, CDM) and discrete diffusion models (e.g., VQ-Diffusion).


**Summary Of The Review:**

Overall, I think the idea is interesting and can be useful for other discrete diffusion models, and the insights into understanding the improving the non-autoregressive transformer methods from the discrete diffusion model perspective could also inspire the better development of non-autoregressive transformers. The experiments are well-executed and relatively strong (although I have the concern about the larger computational cost due to the extra corrector steps). Thus, my initial recommendation is an accept.

---

> ### Author Response · Authors · 2022-11-17
> **Thank you for your review.**
>
> Thank you for your review. We address your three concerns below.
>
> 1.    In terms of the quality-computational cost trade-off, our experiments show that the sample quality achieved by DPC cannot be achieved with the baseline parallel sampling method of MaskGIT, even at the same computational cost. For example, in the ImageNet 512x512 benchmark,  MaskGIT with 64 decoding steps (64 NFEs,  similar to DPC-light(5) which uses 66), obtains FID 7.53  and Inception Score (IS) 133.7, versus FID 6.09, IS 228.1 for DPC-light(5). We will add this result in Table 1.
> Moreover, MaskGIT performance appears to saturate with the number of decoding steps, e.g. as shown in the table below, MaskGIT with 128 steps yields FID 7.33, IS 136.9 as compared to FID 6.09 and IS 228.1 (For each setting we did a grid search of the sampling temperature for best FID.)
> | Method | NFE | FID | IS|
> |---------|-----|-----|---|
> |MaskGIT | 64 | 7.53 | 133.7|
> |MaskGIT |128|  7.33| 136.9|
> | DPC-light(5) | 66 | **6.09** | **228.1**|
>
>
>
> 3. Indeed, in the efficient implementation that uses the shortcut time transition, the predictor step is implicit when doing
> $\hat{x}\_0 \sim  \tilde{p}\_\theta (\hat{x}\_0|x\_t)$.
>  We do not directly model the predictor step $ p\_\theta(x\_t | x_{t+1}) $, so we cannot perform it separately. We found this modeling decision to produce good efficiency and reasonable quality.
>
> 4. Thank you for your suggestion. We will add background on Predictor-Corrector sampling to the revision.

---

> > ### Comment · Reviewer_spPv · 2022-12-13
> > **Thank you**
> >
> > Thank you for your response. My concerns have been well addressed and I suggest the authors add the above experiments and explanations to the final version of the paper.

---

### Decision · Program_Chairs · 2023-01-20

**Decision:**

Accept: poster

**Justification For Why Not Higher Score:**

It's main weakness are the choice of model Plackett-Luce not being well justified and an approximation because of hard marginalization that's relegated to the appendix. The predictor corrector also has added computational cost.

**Justification For Why Not Lower Score:**

The papers strengths are that it is a natural extension based on ideas from continuous generation. The results look reasonable overall. It's relatively simple and easy to implement.

**Metareview: Summary, Strengths And Weaknesses:**

The paper generalizes the predictor-corrector to discrete sampling problems. Predictor corrector samplers were developed to sample from diffusions, where instead of sampling at different noise levels sequential a step of langevin MCMC is added to correct the sample before diffusing it. The motivating use for the discrete version it to correct parallel sampling in non-autoregressive transformers or discrete diffusions. The langevin MCMC doesn't apply directly in the discrete case, so the authors suggest using a learned MCMC corrector. The discrete corrector are masks parameterized by the Plackett-Luce model. The corrector is trained to match detailed balance between the generator and the corrector up to a simplified loss. All the reviewers were quite positive before the author response. The author replies cleared up a few questions such as adding background on the predictor corrector.

The papers strengths are that it is a natural extension based on ideas from continuous generation. The results look reasonable overall.

It's main weakness are the choice of model Plackett-Luce not being well justified and an approximation because of hard marginalization that's relegated to the appendix. The predictor corrector also has added computational cost.

**Note From Pc:**

if the above contains the word "oral" or "spotlight" please see: "oral" presentation means -> notable-top-5% and "spotlight" means -> notable-top-25%. As stated in our emails, we are disassociating presentation type from AC recommendations